

# COVID-19 incidence and mortality in Nigeria: gender based analysis

Olubukola O. Olusola-Makinde[1] and Olusola S. Makinde[2]

[1] Department of Microbiology, Federal University of Technology, Akure, Akure, Ondo, Nigeria
[2] Department of Statistics, Federal University of Technology, Akure, Akure, Ondo, Nigeria

## ABSTRACT

**Background:** Coronavirus Disease 2019 (COVID-19) has been surging globally. Risk strata in medical attention are of dynamic significance for apposite assessment and supply distribution. Presently, no known cultured contrivance is available to fill this gap of this pandemic. The aim of this study is to develop a predictive model based on vector autoregressive moving average (VARMA) model of various orders for gender based daily COVID-19 incidence in Nigeria. This study also aims to proffer empirical evidence that compares incidence between male and female for COVID-19 risk factors.

**Methods:** Wilcoxon signed-rank test is employed to investigate the significance of the difference in the gender distributions of the daily incidence. A VARMA model of various orders is formulated for the gender based daily COVID-19 incidence in Nigeria. The optimal VARMA model is identified using Bayesian information criterion. Also, a predictive model based on univariate autoregressive moving average model is formulated for the daily death cases in Nigeria. Fold change is estimated based on crude case-fatality risk to investigate whether there is massive underreporting and under-testing of COVID-19 cases in Nigeria.

**Results:** Daily incidence is higher in males on most days from 11 April 2020 to 12 September 2020. Result of Wilcoxon signed-rank test shows that incidence among male is significantly higher than female ($p$-value $< 2.22 \times 10^{-16}$). White neural network test shows that daily female incidence is not linear in mean ($p$-value $= 0.00058746$) while daily male incidence is linear in mean ($p$-value $= 0.4257$). McLeod-Li test shows that there is autoregressive conditional heteroscedasticity in the female incidence (Maximum $p$-value $= 1.4277 \times 10^{-5}$) and male incidence (Maximum $p$-value $= 9.0816 \times 10^{-14}$) at 5% level of significance. Ljung-Box test (*Tsay, 2014*) shows that the daily incidence cases are not random ($p$-value=0.0000). The optimal VARMA model for male and female daily incidence is VARMA (0,1). The optimal model for the Nigeria's daily COVID-19 death cases is identified to be ARIMA (0,1,1). There is no evidence of massive underreporting and under-testing of COVID-19 cases in Nigeria.

**Conclusions:** Comparison of the observed incidence with fitted data by gender shows that the optimal VARMA and ARIMA models fit the data well. Findings highlight the significant roles of gender on daily COVID-19 incidence in Nigeria.

Corresponding author
Olubukola O. Olusola-Makinde,
ooolusola-makinde@futa.edu.ng

## INTRODUCTION

The end of 2019 saw the rise of a new disease which originated from Wuhan metropolis in China. World Health Organization (WHO) named the disease; novel coronavirus disease 2019 (COVID-19) and, by March 2020, it had become a pandemic (*Purcell & Charles, 2020*; *Sohrabi et al., 2020*). The responsible virus is called severe acute respiratory syndrome coronavirus 2 (SARS-CoV-2) (*Zhu et al., 2020*; *Gorbalenya et al., 2020*), which shares about 80% genetic similarity with Middle East respiratory syndrome coronavirus and the SARS-CoV (*Drosten et al., 2003*).

Some patients of this novel disease, COVID-19, develop moderate and life-threatening conditions with symptoms such as acute respiratory distress syndrome, acute respiratory failure, coagulopathy, septic shock and metabolic acidosis (*Jin et al., 2020*). Due to potentiality of this disease to be severe and fatal, there is need to identify its risk factors especially for life-threatening conditions. This is to ascertain the specific medical and demographic physiognomies with more accuracy; it will equally enable the apposite supportive hands and rapid intensive care unit treatment when required.

In USA, as at 7 August 2020, the total number of confirmed cases was 3,404,410 and 118,145 mortality cases. Of the 118,145 deceased patients, 53% were males, while some other countries such as Peru, India, Burkina Faso, Guatemala, Kenya, Pakistan, Afghanistan, Thailand, Bangladesh and Nepal had mortality rates of males as high as 70%, 73%, 74%, 74%, 74%, 74%, 76%, 76%, 77% and 81% respectively (*UNWOMEN, 2020*). Even though these data may be underreported, *Lau et al. (2020)* employed mortality rate as yardstick for data reliability. Meanwhile, more disparity in gender distribution of COVID-19 incidence has been documented. *Sohrabi et al. (2020)*, *Del Rio & Malani (2020)* and *Cascella et al. (2020)* reported that severe manifestations are more common in males, particularly in the elderly.

Nigeria confirmed its index case in Lagos State on 27 February 2020 (*Nigeria Centre for Disease Control (NCDC), 2020a*). Thereafter, the first 32 cases were managed in Mainland Hospital, Yaba, Lagos State. Out of the 32 cases, 66% were reported to be males (*Bowale et al., 2020*). As of 6 September 2020, the total number of confirmed cases had risen to 54,905 with 64% being males and 36% females (*Nigeria Centre for Disease Control (NCDC), 2020b*). This raises the question if it is a COVID-19 risk factor to be male?

With focus on Nigeria data, this study considers daily incidence of COVID-19 reported cases based on gender. It answers the question whether daily COVID-19 incidence in Nigeria is gender-skewed, as well as provides relationship between daily incidence and gender. A predictive model based on vector autoregressive moving average model is formulated for the gender based daily incidence in Nigeria. Similarly, a predictive model based on univariate autoregressive integrated moving average (ARIMA) model is formulated for the daily death cases in Nigeria. The study targets the optimal fitting model for predictive and inferential purposes.

Recent literature showed the relationship between countries' population and daily incidence cases among some African countries (*Makinde et al., 2020b*). The study also highlighted significant monotone trends in the daily COVID-19 incidence and mortality

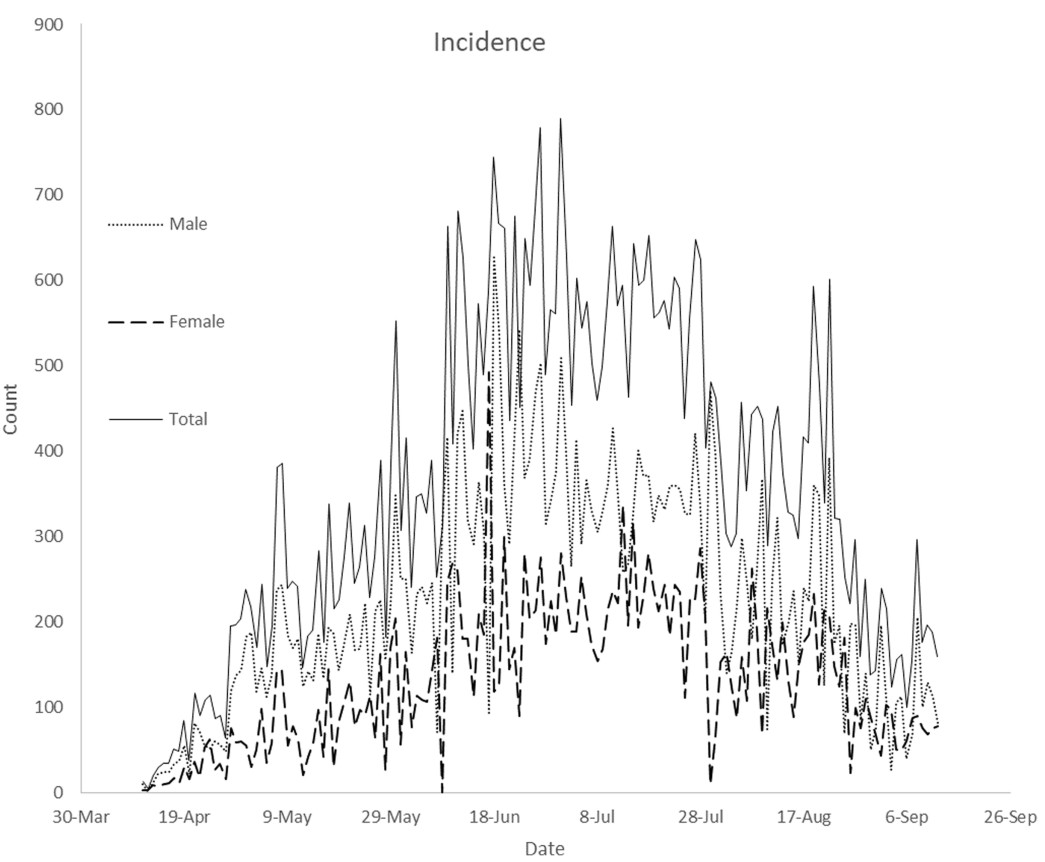

**Figure 1** **Plot of Nigeria's daily COVID-19 incidence by gender from April 11, 2020 to September 12, 2020.** The plot displays compares daily male, female and total incidence from April 11, 2020 to September 12, 2020  

counts of many countries in Africa up to 25 April 2020. *Makinde et al. (2020a)* highlighted effect of modes of transmission of COVID-19 on prevalence and mortality counts in WHO regions. *Ojokoh et al. (2020)* considered the impact of COVID-19 and lockdown policies on farming, food security and agribusiness in some West African countries. Some predictive models have been suggested in literature for modelling COVID-19 dataset. These include use of ARIMA model (*Benvenuto et al., 2020*). *Abdulmajeed, Adeleke & Popoola (2020)* considered an ensemble model based on ARIMA model, a Holt-Winters exponential smoothing model and generalized autoregressive conditional heteroscedasticity model for incidence cases in Nigeria.

## MATERIALS AND METHODS

### Data

The Nigeria's COVID-19 incidence data by gender reported in this study have been sourced from the Nigeria Center for Disease Control (NCDC). The data is available at https://www.ncdc.gov.ng/diseases/sitreps. The data include Nigeria's daily COVID-19 incidence broken down by sex and daily death cases. Figure 1 presents plot of Nigeria's daily COVID-19 incidence by gender from 11 April 2020 to 12 September 2020.

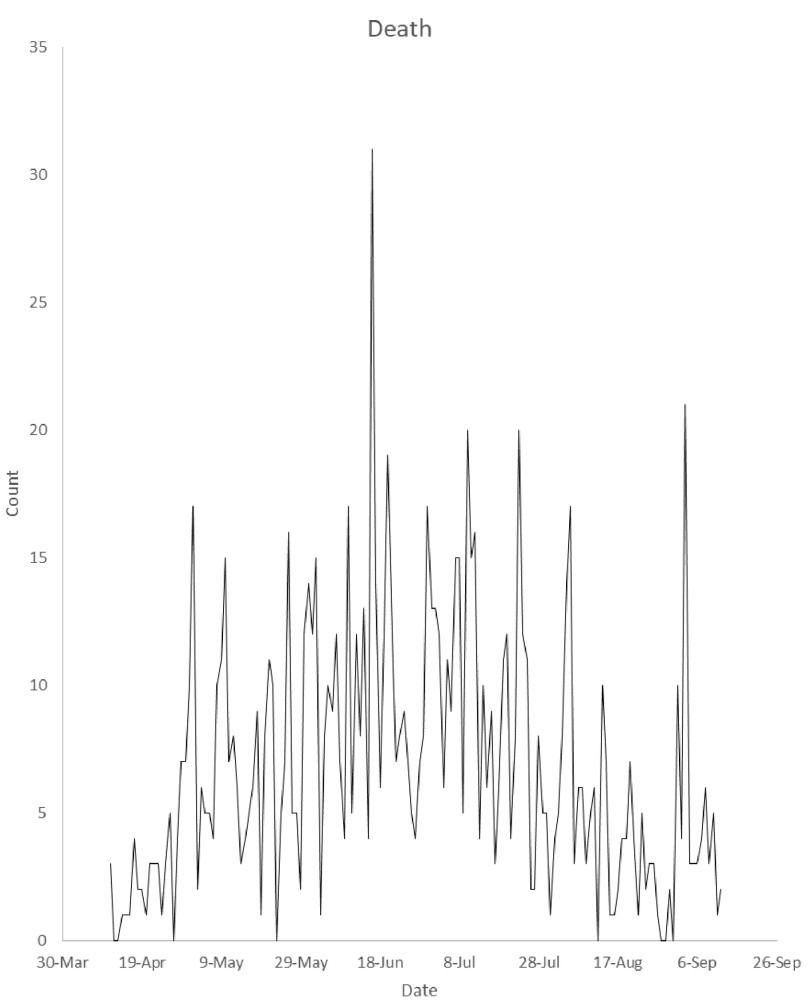

**Figure 2 Plot of number of Nigeria's daily COVID-19 death cases from 11 April 2020 to 12 September 2020.** The plot displays daily COVID-19 death cases from April 11, 2020 to September 12, 2020.

It is observed from the figure that the daily incidence is higher in males in most of the days under study. Daily total incidence was on the rise until the end of July. Figure 2 presents plot of Nigeria's daily COVID-19 death cases by gender from 11 April 2020 to 12 September 2020. The highest daily death cases were recorded on 16 June 2020.

## Dealing with missing values

The data were characterized by eight missing values. The proportion of the original data values that were missing is 0.0516 for male and female daily incidence cases. For estimating missing values, multivariate imputation by chained equations (MICE) (*Van Buuren & Groothuis-Oudshoorn, 2011*) based on random forest was implemented in this study. MICE was applied in *Abiodun et al. (2019)* to estimate a missing daily malaria count and missing values of some climate variables. MICE estimates missing values for continuous data using predictive mean matching approach and binary data using logistic regression (*Abiodun et al., 2019*).

## Multivariate Ljung-Box test

Multivariate Ljung-Box (MLB) test (*Tsay, 2014*) can be employed to investigate whether the daily incidence cases $(Z_t)$ are autocorrelated or random. The MLB test is a multivariate generalisation of the univariate Ljung-Box test. The test rejects the null hypothesis that the daily incidence cases $(Z_t)$ are random if $p$-value of the test is less than a chosen level of significance.

## Augment Dickey–Fuller test

Augmented Dickey–Fuller (ADF) test (*Dickey & Fuller, 1979*) is a statistical test for finding out if a time series contains a unit root. A time series has a unit root implies that the time series is not stationary. The null hypothesis for the ADF test is that a time series has a unit root. A non-stationary time series is differenced d times until it attains stationarity.

## Lagrange multiplier test

Lagrange Multiplier (LM) test (*Tsay, 2014*) was developed to investigate whether a vector time series contains conditional heteroscedasticity. The LM test can be employed to test for autoregressive conditional heteroscedasticity effects in the daily COVID-19 incidence. The test rejects the null hypothesis that there are autoregressive conditional heteroscedasticity effects in the daily COVID-19 incidence if $p$-value of the test is less than a chosen level of significance.

## Vector autoregressive moving average model of order *p* and *q*

Vector autoregressive moving average model of order $p$ and $q$, denoted by VARMA $(p, q)$, is a multivariate generalization of univariate autoregressive moving average model for stationary time series. Mathematically, the VARMA $(p, q)$ model is defined as

$$Z_t = \mu_t + \Phi_1 Z_{t-1} + \Phi_2 Z_{t-2} + \ldots + \Phi_p Z_{t-p} + \varepsilon_t + \theta_1 \varepsilon_{t-1} + \theta_2 \varepsilon_{t-2} + \ldots + \theta_q \varepsilon_{t-q}$$

where $Z_t = \begin{pmatrix} M_t \\ F_t \end{pmatrix}$ is a vector of male and female incidence at time $t$, $\mu_t = \begin{pmatrix} \mu_{1,t} \\ \mu_{2,t} \end{pmatrix}$ is an intercept vector, $\varepsilon_t = \begin{pmatrix} \varepsilon_{1,t} \\ \varepsilon_{2,t} \end{pmatrix}$ is a random error vector, $\Phi_i = \begin{pmatrix} \phi_{i,11} & \phi_{i,12} \\ \phi_{i,21} & \phi_{i,22} \end{pmatrix}$, $i = 1, 2, \ldots, p$ is a parameter of vector autoregressive part and $\theta_i = \begin{pmatrix} \theta_{i,11} & \theta_{i,12} \\ \theta_{i,21} & \theta_{i,22} \end{pmatrix}$, $i = 1, 2, \ldots, q$ is a parameter of vector moving average part. The random error vectors $\varepsilon_t, \varepsilon_{t-1}, \ldots, \varepsilon_{t-q}$ are independently, identically, and normally distributed with mean zero and covariance matrix $\Sigma = E(\varepsilon_t' \varepsilon_t) = \begin{pmatrix} \sigma_{11} & \sigma_{12} \\ \sigma_{21} & \sigma_{22} \end{pmatrix}$. When time series $(Z_t)$ in not stationary, the difference operator $\Delta^d$ is employed on $Zt$ to achieve stationarity. The difference operator $\Delta^d$ is defined as

$$\Delta^d(B) = \begin{pmatrix} (1-B)^{d_1} & 0 & \ldots & 0 \\ 0 & (1-B)^{d_2} & \ldots & 0 \\ \vdots & \vdots & \ddots & \vdots \\ 0 & 0 & \ldots & (1-B)^{d_k} \end{pmatrix}.$$

The coefficients of the VARMA model are estimated using conditional maximum likelihood approach (*Tsay, 2014*).

**The univariate autoregressive integrated moving average model**

The univariate autoregressive integrated moving average model, denoted by ARIMA ($p$, $d$, $q$), on a Nigeria daily reported death cases ($X_t$) at time $t$ is defined as

$$\Delta^d X_t = c + \phi_1 \Delta^d X_{t-1} + \phi_2 \Delta^d X_{t-2} + \ldots + \\ \phi_p \Delta^d X_{t-p} + \varepsilon_t + \theta_1 \varepsilon_{t-1} + \theta_2 \varepsilon_{t-2} + \ldots + \theta_q \varepsilon_{t-q}$$

where $p$, $d$ and $q$ are orders of autoregressive, differenced and moving average parts respectively, $\varepsilon_t$ is the residual of the estimated $Y_t$, which is assumed uncorrelated. $\Delta$ is the difference operator, $\phi_1$, $\phi_1$, $\ldots$, $\phi_p$ are the parameters of the autoregressive part of the model, $\theta_1$, $\theta_2, \ldots, \theta_q$ are parameters of the moving average part of the model. The choice of optimal values of $p$ and $q$ are based on the ARIMA ($p$, $d$, $q$) model with the least Bayesian information criterion. The coefficients of the ARIMA model are estimated using maximum likelihood estimation.

**Case-fatality risks of COVID-19 and fold change**

The crude case-fatality risk (cCFR) (*Lau et al., 2020*) of COVID-19 infections at date t is defined as the ratio of the total number of deaths on day t to the total number of confirmed cases on day $t$. *Lau et al. (2020)* defined an cCFR-adjusted total cases of country A at date t relative to country B as

$$\text{ATCC}(A) = \text{total reported cases}(A) \times \frac{\text{cCFR}(A)}{\text{cCFR}(B)}$$

where ATCC ($A$) is the cCFR-adjusted total COVID-19 cases of country A at date $t$, cCFR ($A$) and cCFR ($B$) are crude case-fatality risk of countries $A$ and $B$ respectively at time $t$. The fold change at date $t$ is defined as

$$\text{fold change at date } t = \frac{\text{cCFR} - \text{adjusted total cases at date } t}{\text{total reported cases at date } t}.$$

## RESULTS

There is a significantly increasing trend ($p$-value $< 2.22 \times 10^{-16}$) in daily male incidence from 11 April 2020 to 18 June 2020 while there is a significantly decreasing trend ($p$-value $= 1.9091 \times 10^{-12}$) in daily male incidence from 19 June 2020 to 12 September 2020. There is a significantly increasing trend ($p$-value $< 2.22 \times 10^{-16}$) in daily female incidence from 11 April to 13 July 2020 while there is a significantly decreasing trend ($p$-value $= 1.4916 \times 10^{-6}$) in daily female incidence from 14 July 2020 to 12 September 2020. Similarly, there is a significantly increasing trend ($p$-value $< 2.22 \times 10^{-16}$) in daily total incidence from 11 April 2020 to 1 July 2020 while there is a significantly decreasing trend ($p$-value $= 2.084 \times 10^{-12}$) in daily total incidence from 2 July 2020 to 12 September 2020. There is a significantly increasing trend ($p$-value $= 3.5763 \times 10^{-7}$) in

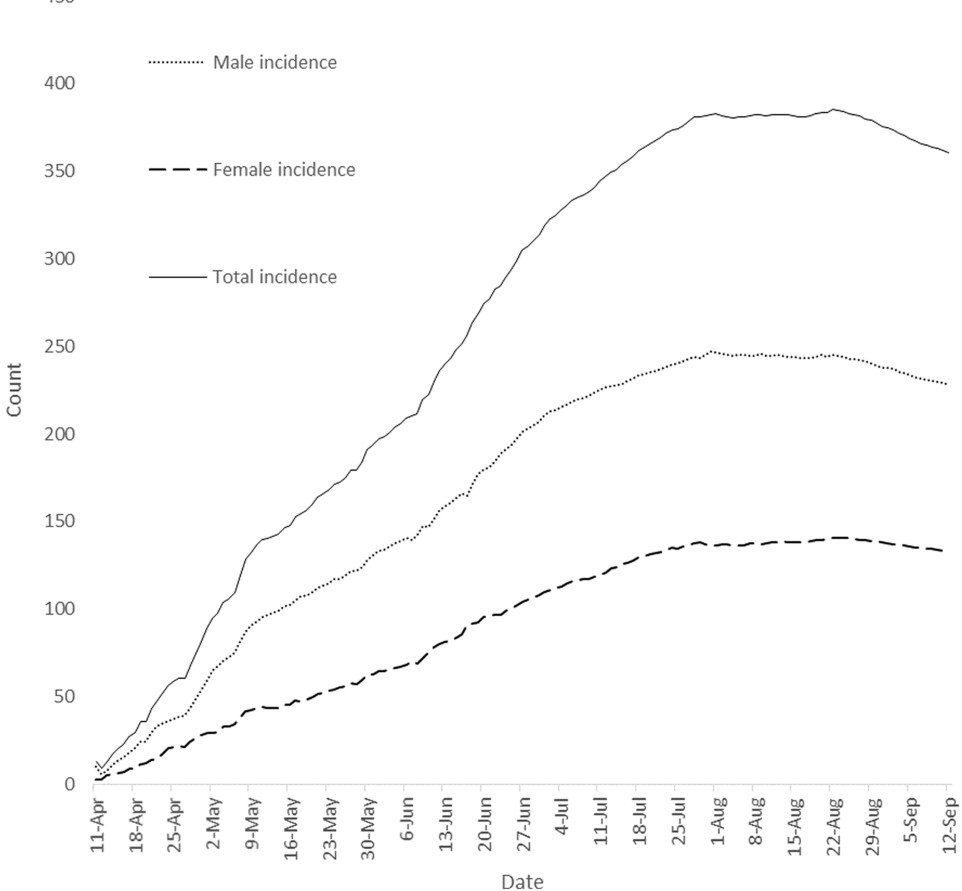

**Figure 3 Plot of daily mean incidence over time.** The plot displays how means of daily male, female and total incidence vary from April 11, 2020 to September 12, 2020.

daily death cases from 11 April 2020 to 16 June 2020 while there is a significantly decreasing trend ($p$-value = $2.1484 \times 10^{-14}$) in daily death cases from 17 June 2020 to 12 September 2020.

White-neural-network test shows that daily female incidence is not linear in mean ($p$-value = 0.00058746) while daily male incidence is linear in mean ($p$-value = 0.4257). McLeod-Li test was applied to investigate whether there are no autoregressive conditional heteroscedasticity effects in the daily incidence cases. There is an autoregressive conditional heteroscedasticity effect in daily female incidence if the mean daily female incidence changes over time. This implies that on the average, the daily female incidence changes significantly over time. This is shown in Fig. 3. The McLeod-Li test shows that there is autoregressive conditional heteroscedasticity in the female incidence (Maximum $p$-value = $1.4277 \times 10^{-5}$) and male incidence (Maximum $p$-value = $9.0816 \times 10^{-14}$) at 5% level of significance. This is confirmed by the LM test for vector of male and female incidence. The LM test shows that there are autoregressive conditional heteroscedasticity

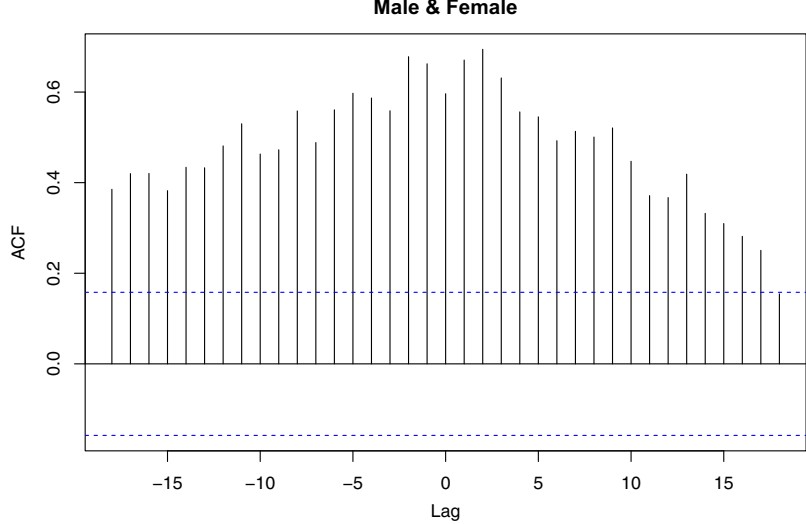

**Figure 4 Plot of cross correlation function for daily male and female COVID-19 incidence in Nigeria.** The plot displays how male incidence at previous days affect female incidence and vice versa.

**Table 1 Augmented Dickey–Fuller test of stationarity for Nigeria's daily COVID-19 incidence.**

|  | Male | Female |
|---|---|---|
| Dickey–Fuller test statistic | −1.3750 | −1.4331 |
| $p$-Value | 0.8369 | 0.8127 |

effects in the daily COVID-19 incidence ($p$-value = 0.0001). Similarly, the multivariate Ljung-Box test (*Tsay, 2014*) shows that the daily incidence cases ($Zt$) are not random ($p$-value = 0.0000). This implies that the daily incidence cases are autocorrelated. Figure 4 presents the plot of cross correlation between the daily male and female incidence at $k$ previous days. It was observed that the cross correlation values are positive at all lags and significant up to 18 previous days. This implies that daily female incidence at day $t$ is significantly increasing as the daily male incidence increases at day $t - k$.

The Wilcoxon signed-rank test was employed to test if the null hypothesis that confirmed male and female cases have the same distribution against the alternative hypothesis that the distributions of male and confirmed female cases are not the same. Result of the test shows that confirmed cases in male and female have different distributions ($p$-value $< 2.22 \times 10^{-16}$) and the number of confirmed cases is significantly higher in males than in females.

Augment Dickey–Fuller test was employed on each of the series to investigate if the confirmed male and female incidence cases are stationary at 5% level of significance. The test rejects the hypothesis that a time series is not stationary if $p$-value of the test is less than 0.05. Table 1 presents the test statistics and p-values of the test for incidence by gender. The Nigeria's daily COVID-19 incidence is not stationary for male ($p$-value = 0.9328) and female patients ($p$-value = 0.9069).
**Table 2 Estimates of coefficients of optimal VARMA (0, 1) models.**

| Coefficients | Estimate | Std. error | t Value | Pr(>|t|) |
|---|---|---|---|---|
| $\mu_{1,t}$ | 0.484 | $9.24 \times 10^{-6}$ | 52,353 | $< 2 \times 10^{-16}$*** |
| $\mu_{2,t}$ | −0.453 | $8.61 \times 10^{-6}$ | −52,670 | $< 2 \times 10^{-16}$*** |
| $\theta_{1,11}$ | −0.953 | $5.91 \times 10^{-6}$ | −161,193 | $< 2 \times 10^{-16}$*** |
| $\theta_{1,12}$ | 0.348 | $2.36 \times 10^{-6}$ | 147,627 | $< 2 \times 10^{-16}$*** |
| $\theta_{1,21}$ | 0.101 | $9.44 \times 10^{-7}$ | 106,724 | $< 2 \times 10^{-16}$*** |
| $\theta_{1,22}$ | −0.861 | $5.08 \times 10^{-6}$ | −169,620 | $< 2 \times 10^{-16}$*** |

**Note:**
*** implies significance at 0.0001.

The difference operation of the daily COVID-19 incidence was performed. The data was differenced once to attain stationarity, that is $\Delta^d(B) = \begin{pmatrix} 1-B & 0 \\ 0 & 1-B \end{pmatrix}$. Then, VARMA $(p, q)$ model of various orders were fitted to the differenced data with the aim of identifying the optimal VARMA model for the data.

The optimal VARMA model is chosen as the model that minimises the Bayesian information criterion. The optimal model is VARMA (0, 1) model. Table 2 presents the estimates of coefficients of optimal VARMA (0, 1) model. The optimal VARMA (0, 1) model can be expressed mathematically as:

$$Z_t - Z_{t-1} = \mu_t + \varepsilon_t + \theta_1 \varepsilon_{t-1}$$

where

$$\mu_t = \begin{pmatrix} 0.4836 \\ -0.4532 \end{pmatrix}, \quad \theta_1 = \begin{pmatrix} -0.953 & 0.348 \\ 0.101 & -0.861 \end{pmatrix},$$

and the covariance matrix of residuals is $\Sigma = \begin{pmatrix} 5166.7336 & -187.7683 \\ -187.7683 & 3378.1531 \end{pmatrix}$.

The optimal VARMA (0, 1) model can also be presented as two univariate regression models:

$$M_t - M_{t-1} = 0.484 + \varepsilon_t - 0.953\varepsilon_{t-1}$$

$$F_t - F_{t-1} = -0.453 + \varepsilon_t - 0.861\varepsilon_{t-1}$$

The test of significance of estimates of coefficients of the models shows that estimates of coefficients of the models are significantly different from zero ($p$-value $< 0.05$). This is given in Table 2. The implication of the above equations is that daily male incidence does not significantly depend on female incidence at previous days. Similarly, number of female incidence does not depend on male incidence at previous days. Figure 5 presents the comparison of the observed incidence with fitted data by gender. It can be observed from the figure that the models fit the data well.

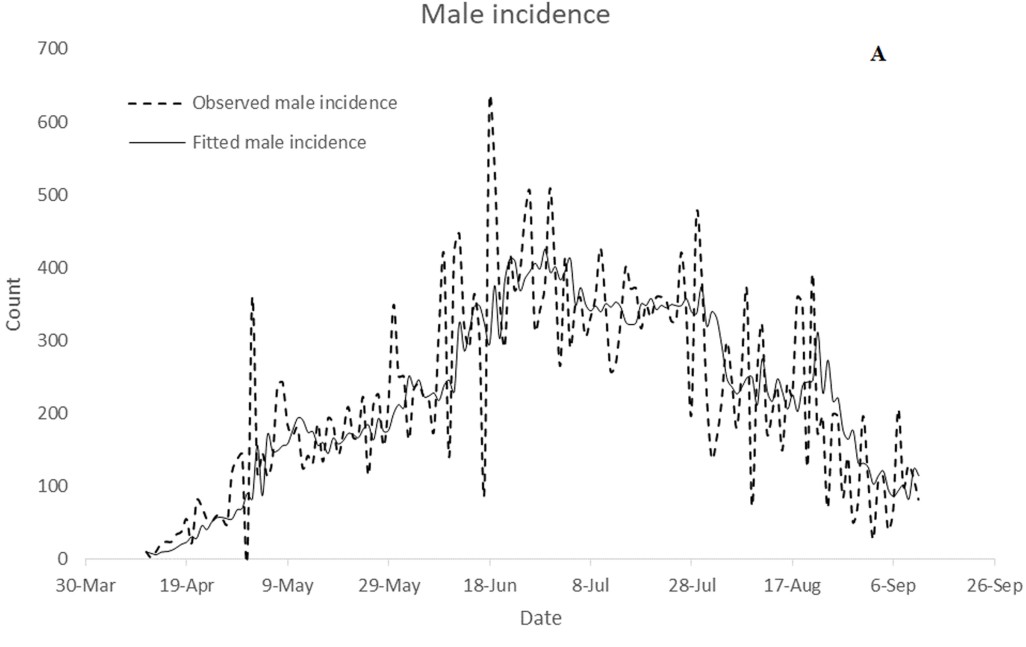

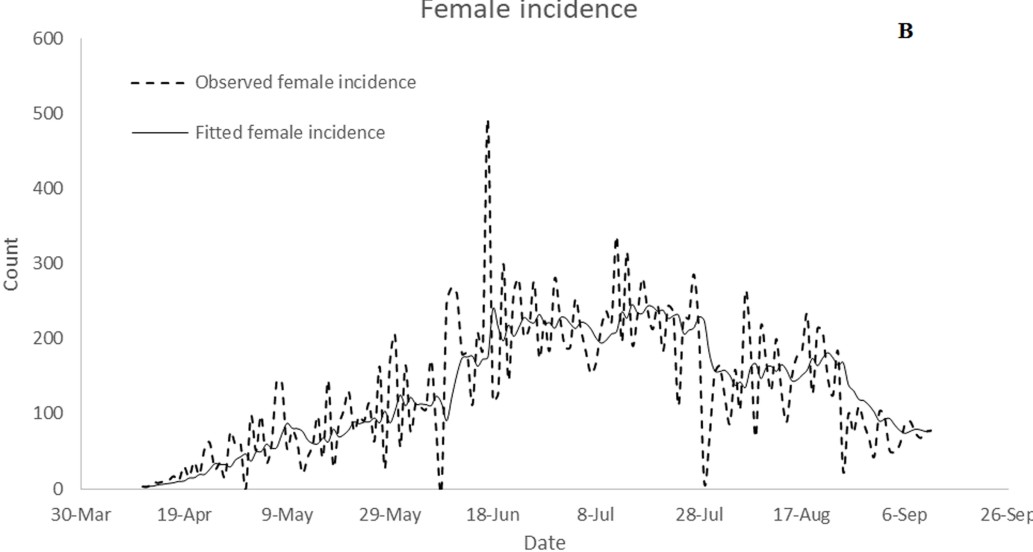

**Figure 5 Comparison of observed data with fitted confirmed incidence: (A) male and (B) female.**

The calculated cCFR value of Nigeria is 0.0192. The total reported cases as of 12 September 2020 is 440,248. The calculated cCFR value of Germany as of 12 September 2020 is 0.0362.

The total reported cases as of 12 September 2020 is 440,248 while the estimated total COVID-19 incidence cases based on cCFR is 233,188. The fold change value of Nigeria is estimated to be 0.5297.

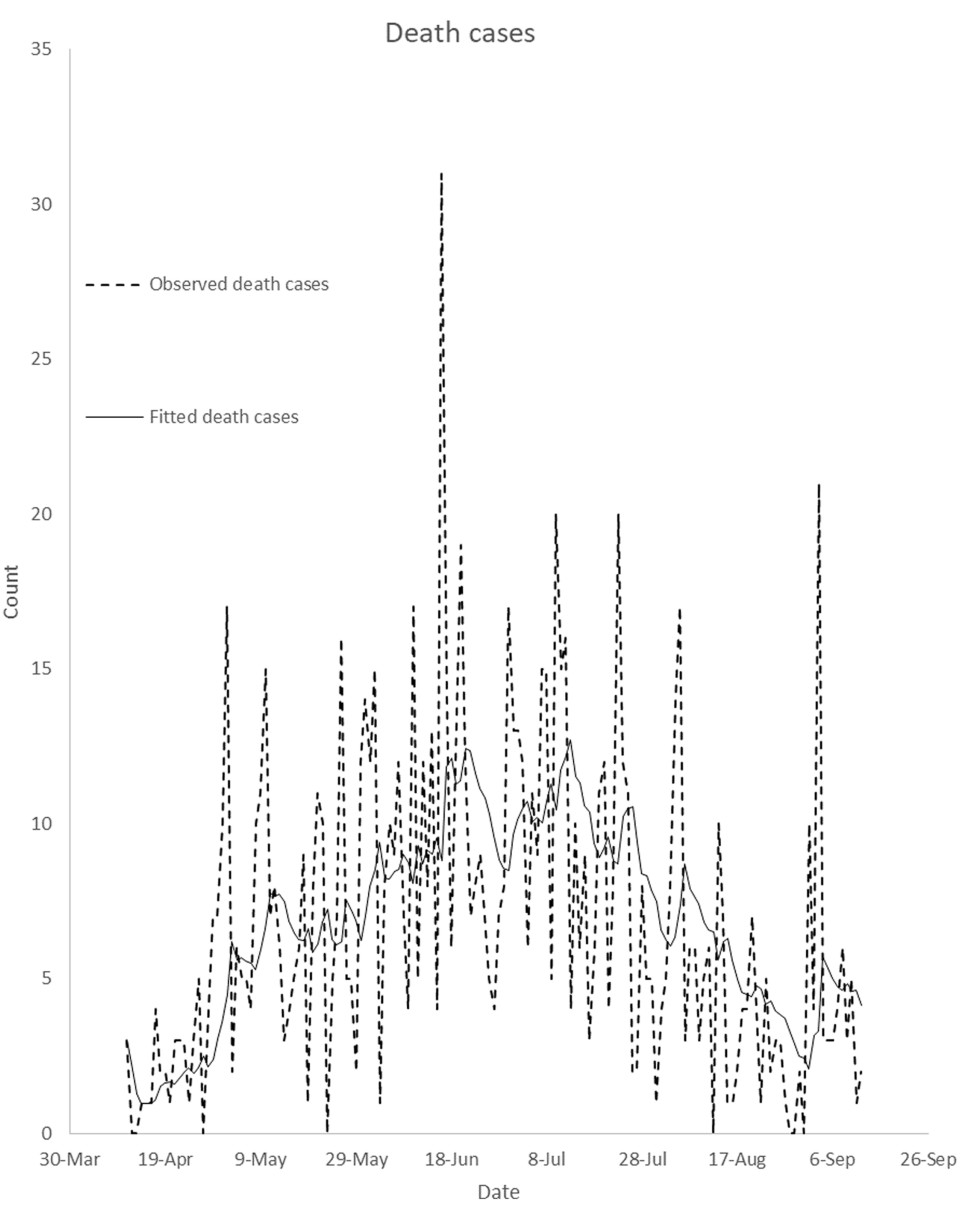

**Figure 6 Comparison of observed data with fitted death cases in Nigeria.** It compares observed daily death cases with daily fitted death cases from April 11, 2020 to September 12, 2020. This is to illustrate how well the optimal ARIMA model fits the observed death cases.

## Modelling Nigeria's daily COVID-19 death cases

Augmented Dickey–Fuller test was employed to investigate the stationarity of Nigeria's daily COVID-19 death cases. The ADF test ($p$-value = 0.0888) fails to reject the null hypothesis that the daily reported death cases are stationary at 5% level of significance. This implies that the daily COVID-19 death cases are not stationary. That is, the average and variance of reported death cases change with time. To overcome this, a univariate autoregressive moving average (ARIMA) model of various orders was formulated for Nigeria's daily COVID-19 death cases. Optimal model was selected following *Makinde &*

*Fasoranbaku (2011)* and *Benvenuto et al. (2020)*. The optimal model for the Nigeria's daily COVID-19 death cases is identified to be ARIMA (0, 1, 1). Figure 6 presents the comparison of the observed death cases with fitted data. It can be observed from the figure that the optimal ARIMA (0, 1, 1) model fits the data well.

## DISCUSSION

Daily confirmed cases in Nigeria are gender-skewed (Fig. 1). The daily male-female ratios of COVID-19 confirmed cases are higher than one over the study period. This is confirmed in Fig. 3. In the figure, daily average confirmed cases are consistently higher in males but monotonically increasing till 8 August 2020. Result of Wilcoxon signed-rank test shows that confirmed male cases are not distributed equivalently as confirmed female cases. The result of the test confirms that confirmed male and female cases have different distributions and the number of confirmed cases is significantly higher in males. *Bianconi et al. (2020)* argued that gender inequalities may have influenced either the estimation of COVID-19 cases or the susceptibility to COVID-19 over time. *Bianconi et al. (2020)* observed a time-related changes in sex distribution of COVID-19 cases in Italy as well as declining infection proportion with male-female ratio from 9 March to 11 May 2020. However, there is no significant time-related changes in sex distribution of COVID-19 cases in Nigeria. *Bianconi et al. (2020)* argued that males have been reported to be at higher risk of developing symptoms and severe clinical manifestations of COVID-19, and may have had a higher access to COVID-19 diagnostic tests at the beginning of the observed period as compared to females.

*Gebhard et al. (2020)* claimed that COVID-19 is deadlier for infected men than women in China in terms of fatality rate, and that differential susceptibility between males and females may be imparted in diverse ways by gender for infectious diseases caused by viruses. *Klein & Huber (2009)* argued that susceptibility and response to viral infections, incidence and disease severity may differ in males and females. *Gadi et al. (2020)* attributed higher risks of COVID-19 in males to the lack of the stimulatory effects of estrogen and that androgens which males produce seem to have a defensive mechanism against their immune response. *Jin et al. (2020)* investigated the role of gender in morbidity and mortality in patients with COVID-19 in China and concluded that men could be more at risk of death by COVID-19.

The confirmed male and female incidence cases were not stationary. This implies that either their means, variances or covariances increased with time (Fig. 3). The confirmed male and female incidence cases were differenced once in order to achieve stationarity. The VARMA model was fitted to the differenced data. The fitted VARMA model showed that daily male incidence does not significantly depend on female incidence at previous days. Similarly, number of female incidence does not depend on male incidence at previous days. Comparison of the observed incidence with fitted data showed that the models fit the data well (Fig. 5). Similarly, ARIMA (0, 1, 1) model was selected as the optimal ARIMA ($p$, $d$, $q$) for fitting daily reported death cases using Bayesian information criterion. The model fits the data well, as shown in Fig. 6. This is in line with study of *Benvenuto et al. (2020)*, where ARIMA models were performed on the Johns Hopkins

epidemiological data to predict the epidemiological trend of the prevalence and incidence of COVID-19. However, parameters of optimal ARIMA models were not identified in their study.

In evaluating massive under-reporting and under-testing of total reported COVID-19 cases in some global epicenters, cCFR value of Germany was used as a standard in *Lau et al. (2020)*. In this study, cCFR value of Germany was also considered as the standard. As of 12 September 2020, the cCFR values of Germany, Greece and South Korea are 3.62%, 2.36% and 1.61% respectively while the cCFR value of Nigeria is 1.92%. The estimated total COVID-19 incidence cases is 233,188. There is no evidence of massive under-detection and under-reporting in Nigeria. When estimating the real total amount of COVID-19 cases in Nigeria, no considerable increase in the number of COVID-19 incidence is detected in comparison to Nigeria's total reported COVID-19 cases. Fold change can be used as an indicator for under-reporting and under-detecting COVID-19 cases. Nigeria's fold change value is estimated to be 0.5297 as of 12 September 2020. This value is comparable to fold change of South Korea but less than that of Italy, France, Spain, Iran, China and the USA as of 17 March 2020. This study is limited by dearth of information on mortality cases by gender and daily population in Nigeria.

## CONCLUSIONS

This study provides insight into how COVID-19 affects one sex than the other in Nigeria over a period of time. In particular, a vector autoregressive moving average model was formulated for predicting gender based—daily incidence in Nigeria. Also, a predictive model based on autoregressive integrated moving average model was presented for daily death cases. The findings show that males were at higher risk of contracting COVID-19 than females. The mean daily incidence changes over time for male and female but the change is significant for female individuals. The significance of gender on daily incidence was highlighted. The vector autoregressive moving average model shows that male incidence does not depend on number of female incidence at previous days. Similarly, number of female incidence does not depend on number of male incidence at previous days. Massive underreporting and under-testing was not observed in Nigeria's reported COVID-19 cases based on crude case-fatality risk and fold change.

### Funding
The authors received no funding for this work.

### Competing Interests
The authors declare that they have no competing interests.

### Author Contributions
- Olubukola O. Olusola-Makinde conceived and designed the experiments, performed the experiments, authored or reviewed drafts of the paper, and approved the final draft.

- Olusola S. Makinde performed the experiments, analyzed the data, prepared figures and/or tables, authored or reviewed drafts of the paper, and approved the final draft.

## Data Availability

Raw data showing daily incidence and mortality of COVID-19 in Nigeria, including the gender disparities, are available as a Supplemental File.

## Supplemental Information

Supplemental information for this article can be found online at http://dx.doi.org/10.7717/peerj.10613#supplemental-information.

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
