# Peer review of "COVID-19 incidence and mortality in Nigeria: gender based analysis"

_PeerJ, doi:10.7717/peerj.10613_

## Round 0.1 · original submission · Major Revisions

Dear author, the COVID-19 incidence by sex is interesting, but it is necessary to make some improvements according to the reviewers' recommendations for a good quality manuscript. I soon hope your incorporated improvements.

·

Basic reporting

NO Comment

Experimental design

NO Comment

Validity of the findings

Comments

Manuscript Lines 64 to 67
“Of the 118,145 deceased patients, 53% were males, while some other countries such as Peru, India, Burkina Faso, Guatemaia, Kenya, Pakistan, Afghanistan, Thailand, Bangladesh and Nepal had mortality rates of males as high as 70%, 73%, 74%, 74%, 74%, 74%, 76%, 76%, 77% and 81% respectively (UNWOMEN, 2020).”
Reviewer’s comment:
The gender skew heretofore reported and the subject of present study is based on UNWOMEN. (2020). Emerging Gender Data and Why It Matters, Available from: https://data.unwomen.org/resources/covid-19-emerging-gender-data-and-why-it-matters,(accessed September 6, 2020)

The relevant website mentions prominently the following

"Source: Data submitted to NCOVmart reported through the global surveillance system of WHO, as of 14 July 2020.
Notes: Data cleaning are ongoing and in progress. All numbers should be interpreted with caution. As of 14 July 2020, 12,880,565 cases were reported. Data presented here, therefore, represent only 37% of all reported cases."

It is not denied that there may be a gender skew but the reliability of data or relative lack thereof that forms the basis of such conclusion has to be mentioned prominently in the manuscript.

Moreover there are reports of massive underreporting of COVID19 cases globally “Lau H, Khosrawipour T, Kocbach P, Ichii H, Bania J, Khosrawipour V. Evaluating the massive underreporting and undertesting of COVID-19 cases in multiple global epicentres.. Pulmonology. 2020;S2531-0437(20)30129-X. doi:10.1016/j.pulmoe.2020.05.015”

Therefore the extent of underreporting in different countries has to be mentioned and taken into consideration even if the authors speculate that underreporting is more or less equal for male and female members in a given population.

The authors of the present study do admit to the uncertainty of the nature of data the analysis is based on by noting in Lines 254 and 255 that “This study is limited by dearth of information on mortality cases by gender and daily population in Nigeria.”
However in a manuscript with rigorous statistical methods employed such as in the present study, the authors are expected to quantify the extent of underreporting in a way similar to the one carried out in “Lau H, Khosrawipour T, Kocbach P, Ichii H, Bania J, Khosrawipour V. Evaluating the massive underreporting and undertesting of COVID-19 cases in multiple global epicentres.. Pulmonology. 2020;S2531-0437(20)30129-X. doi:10.1016/j.pulmoe.2020.05.015”

Reviewer’s Conclusion: The authors ought to mention the degree or uncertainty of the data that forms the basis of analysis. The said nature of data ought to be mentioned in statistical terms as opposed to subjective and qualitative.

Additional comments

NO Comment

Reviewer 2 ·

Basic reporting

This manuscript provides a good overview of what its goal is, namely to highlight the gender differences in COVID-19 in Nigeria. It provides sufficient references in the introduction but has no discussion section. The latter leaves a lot to be desired. Differences in COVID-19 incidence by sex have been noted worldwide, but none of that literature is cited.
Figures could be presented much better, with axes not labelled. In fact, Figure 4 does not have a title. The figures need to be strengthened to make it easy for the reader to understand the message. Tables 1 and 2 are not labelled.
Overall, the paper reads like a draft and needs much revision. Grammar and spelling errors abound.

Experimental design

The methods section does a good job of displaying all the tests that are employed but is not exhaustive. For example, Augmented Dickey-Fuller test is not mentioned in the methods but in the results for the first time.

Validity of the findings

The data provided offer scope for replication. I do not have the specialization to fully assess the veracity of the statistics employed in this manuscript.
Limitations of the methods employed are not described in the discussion section, as is typically expected.

Additional comments

Overall, this manuscript is off to a good start and requires significant revision. Grammar and editing for clarity is much needed. All tests performed should be first mentioned in the methods section. I recommend separating the results and discussion sections- once that is done, more relevant literature ought to be cited in the discussion.

Annotated reviews are not available for download in order to protect the identity of reviewers who chose to remain anonymous.

---

## Round 0.2 · accepted · Accept

Dear authors,

I have reviewed and verified that you have made the improvements indicated by the reviewers. I consider that its results, focused on gender cases (most of them are male) are very specific towards the country under study. In these times of pandemic, your contribution is valuable and pertinent, Therefore, we accept for publication.